# Modern Backbones Improve Multi-task DETR for Mammography Classification and Lesion Localization

**Dinh Tan Nguyen**[1,2,†] (iD)                                       DINHTAN.NGUYEN@UTS.EDU.AU
**Quang-Hien Kha**[2,3,4,†] (iD)                                        D142111015@TMU.EDU.TW
**Le-Hoang Nguyen**[3] (iD)                                  LEHOANGNGUYEN510@GMAIL.COM
**Minh-Toan Dinh**[7] (iD)                                       TOANDINH6501@OUTLOOK.COM
**Xuan-Huy Nguyen**[2] (iD)                                          HUYLOP99@GMAIL.COM
**Dac Phu Ho**[1] (iD)                                              C3514490@UON.EDU.AU
**Cao Truong Tran**[6] (iD)                                       TRUONGCT@LQDTU.EDU.VN
**Sai Ho Ling**[1] (iD)                                            STEVE.LING@UTS.EDU.AU
**Lan T Ho-Pham**[3] (iD)                                      LAN.HOPHAM@SAIGONMEC.ORG
**Liem Pham**[2]                                              LIEM.PHAM@SAIGONMEC.ORG
**Nguyen Quoc Khanh Le**[3,4*] (iD)                                  KHANHLEE@TMU.EDU.TW

[1] *University of Technology Sydney, Australia.* [2] *Saigon Precision Medicine Research Center, Vietnam.* [3] *College of Medicine, Taipei Medical University, Taiwan.* [4] *AIBioMed Research Group, Taipei Medical University, Taiwan.* [6] *Le Qui Don Technical University, Vietnam.* [7] *International Graduate Program in Artificial Intelligence, National Central University, Taiwan.* [†] *These authors contributed equally to this work.*

## Abstract

Joint exam-level prediction and candidate-region localization may improve the usefulness of AI support in mammography. We study this setting using a multi-task DETR framework, where shared representations support both image-level malignancy prediction and lesion localization, and evaluate its performance on OPTIMAM and a biopsy-confirmed SGM1k cohort. Across both datasets, modern backbones consistently outperformed older ResNet-style features, with ConvNeXtV2 and DINOv3 giving the strongest overall results, whereas MambaVision was less competitive. On OPTIMAM, ConvNeXtV2 achieved the best overall performance, reaching 97.96% AUC, 99.89% sensitivity, 25.08% mAP@.5, and 74.38% recall@.25. On SGM1k, DINOv3 gave the strongest overall results, with 90.97% AUC, 86.28% sensitivity, 82.00% specificity, 27.04% mAP@.5, and 77.32% recall@.25. These findings suggest that backbone quality is a critical factor in effective multi-task mammography, with ConvNeXtV2 emerging as a particularly strong and well-matched CNN backbone for mammography in this framework.[1]

**Keywords:** Mammography, multi-task learning, object detection, classification, DETR, lesion localization

## 1. Introduction

Screening mammography requires both exam-level risk assessment and spatial evidence that can direct reader attention. This remains challenging because suspicious findings may be

---

1. Code is publicly available at `https://github.com/saigonmec/mammo2detr`.

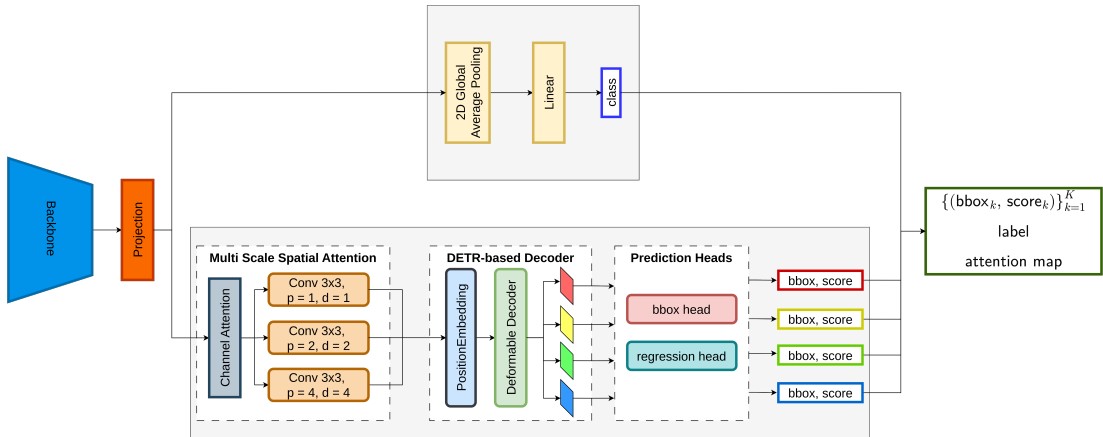

Figure 1: Overview of the proposed multi-task DETR architecture

subtle, small, and partially masked by dense tissue; in a large screening study, mammographic sensitivity dropped substantially in the densest breasts (Kolb et al., 2002). Screening decisions must also balance benefit and harm: a systematic review of breast cancer screening reported a non-trivial cumulative risk of false-positive biopsy findings, especially with more frequent screening (Myers et al., 2015). For AI systems intended as decision support, it is therefore valuable to assess not only classification performance but also whether the model can return plausible candidate regions. Multi-task learning is attractive in this setting because a shared representation can support both image-level malignancy prediction and lesion localization within a single framework (Kha et al., 2024). DETR-style detectors provide an appealing basis for this approach because they predict a set of object instances end to end without hand-crafted anchors (Carion et al., 2020). Here, we study a shared multi-task DETR architecture for joint mammography classification and lesion localization, with a particular focus on backbone suitability. Because the quality of the shared representation is central to multi-task performance, not all backbone families may be equally effective in this setting. We therefore compare ResNet50 (He et al., 2016), ConvNeXtV2-Tiny (Woo et al., 2023), MambaVision-Tiny (Hatamizadeh and Kautz, 2025), and DINOv3 ViT-B/16 (Siméoni et al., 2025) across OPTIMAM (OMI-DB) (Halling-Brown et al., 2020) and a biopsy-confirmed SGM1k cohort (Kha et al., 2024).

## 2. Method

We use a fixed multi-task DETR framework for joint mammography classification and lesion localization. As illustrated in Figure 1, the proposed architecture combines an interchangeable visual backbone with a shared feature projection layer, an image-level classification branch, and a query-based localization branch based on a Deformable DETR-style decoder (Zhu et al., 2020). This design enables a controlled comparison of backbone effects while keeping the downstream prediction heads unchanged across experiments. We evaluated the framework on two mammography datasets, including OPTIMAM (Halling-Brown et al., 2020) and the biopsy-confirmed SGM1k cohort, comprising 24,643 and 3,525 images after preprocessing, respectively. Data preprocessing, training, and evaluation were standardized

Table 1: Classification and localization performance

| Metric | OPTIMAM (OMI-DB) | | | | Oncology Hospital (SGM1k) | | | |
|---|---|---|---|---|---|---|---|---|
| | ResNet50 | ConvNeXtV2 | Mamba | DINOv3 | ResNet50 | ConvNeXtV2 | Mamba | DINOv3 |
| **Classification (%)** | | | | | | | | |
| Acc | 89.37 | 89.99 | 91.71 | **91.94** | 76.37 | 83.71 | 77.84 | **84.25** |
| AUC | 97.36 | **97.96** | 96.92 | 97.35 | 86.62 | 90.44 | 84.74 | **90.97** |
| Sens | 99.78 | **99.89** | 90.62 | 95.35 | **90.70** | 81.40 | 73.26 | 86.28 |
| Spec | 83.00 | 84.00 | 88.00 | **90.00** | 57.00 | **87.00** | 84.00 | 82.00 |
| F1 | 89.54 | 90.15 | 91.83 | **92.02** | 75.45 | 83.79 | 77.96 | **84.25** |
| **Detection (%)** | | | | | | | | |
| IoU | 20.30 | **33.19** | 19.43 | 27.67 | 21.46 | 31.65 | 19.66 | **32.65** |
| mAP@.5 | 18.41 | **25.08** | 12.90 | 18.05 | 11.26 | 23.79 | 12.20 | **27.04** |
| mAP@.25 | 48.72 | **55.38** | 32.27 | 38.08 | 40.80 | 41.65 | 27.58 | **46.00** |
| R@.25 | 63.52 | **74.38** | 52.15 | 67.15 | 64.46 | 72.40 | 55.58 | **77.32** |

across experiments following the pipeline described in Appendices B and C, while additional architectural details are provided in Appendix A.

## 3. Results and Discussion

The results show that the proposed multi-task DETR framework performs best when paired with a suitable modern backbone. Across both datasets, ConvNeXtV2 and DINOv3 achieved the strongest overall performance, whereas ResNet50 was consistently less competitive and MambaVision showed weaker overall results. On OPTIMAM, ConvNeXtV2 gave the best overall metrics, suggesting that an improved CNN backbone is particularly well suited to mammography images. On SGM1k, DINOv3 performed best overall and achieved the strongest localization results, while ConvNeXtV2 again preserved the highest specificity.

From a clinical perspective, the detection metrics in Table 1 should be interpreted as candidate-region support rather than precise lesion delineation. The Grad-CAM visualizations in Appendix D show that stronger backbones more consistently focused on clinically relevant suspicious regions. Such approximate localization may still be useful for directing reader attention in mammography, where small abnormalities and tissue overlap make exact boundaries difficult, particularly in dense breasts (Kolb et al., 2002). Overall, backbone quality is a key determinant of effective multi-task mammography, with modern CNN and ViT representations providing the most suitable shared features for joint classification and localization.

## 4. Conclusion

Our findings suggest that the success of multi-task DETR in mammography depends strongly on representation quality. Modern backbones provided a better balance between exam-level prediction and candidate-region localization, highlighting backbone suitability

as a key design factor in multi-task breast imaging. In particular, ConvNeXtV2 appears especially well matched to the fine-grained visual patterns of mammography.

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

# Appendix A. Architecture Details

## A.1. Overall multi-task design

Given an input mammogram $I \in \mathbb{R}^{3 \times H \times W}$, the model jointly predicts an image-level malignancy label and a set of query-based lesion proposals:

$$f_\theta(I) = \left( \hat{y}, \{(\hat{b}_k, \hat{s}_k)\}_{k=1}^K \right),$$

where $\hat{y} \in \mathbb{R}^C$ denotes the image-level logits for $C$ classes, $\hat{b}_k \in [0,1]^4$ is the $k$-th predicted bounding box in normalized coordinates, and $\hat{s}_k \in [0,1]$ is its corresponding objectness score. The architecture consists of an interchangeable visual backbone, a classification branch for image-level prediction, and a localization branch for lesion proposal generation.

## A.2. Backbone interface

To enable a controlled comparison across heterogeneous backbone families, the backbone output is projected into a common feature representation using a 1×1 convolution:

$$\mathbf{F} = \phi(\mathbf{F}_{\text{backbone}}) \in \mathbb{R}^{B \times 256 \times H' \times W'}.$$

This projection standardizes the channel dimension while keeping the downstream detection and classification heads unchanged across experiments. The compared backbones were ResNet50, ConvNeXtV2-Tiny, MambaVision-Tiny, and DINOv3 ViT-B/16.

## A.3. Classification branch

The classification branch applies global average pooling to the projected feature map $\mathbf{F}$, followed by a learnable linear classifier:

$$\hat{y} = W \cdot \text{GAP}(\mathbf{F}) + b$$

where $W$ and $b$ denote the weights and bias of the final linear layer. This branch produces the image-level logits for malignancy classification.

## A.4. Localization branch

The localization branch applies a lightweight multi-scale spatial module to enhance local lesion cues before decoding. Specifically, three parallel 3×3 convolutions with different dilation rates are used to capture multiple receptive fields:

$$(d, p) \in \{(1,1), (2,2), (4,4)\}.$$

The resulting features are aggregated and passed to a Deformable DETR-style decoder, which uses a fixed set of learned object queries to predict lesion proposals. Each query outputs a bounding box $\hat{b}_k$ and an objectness score $\hat{s}_k$. This design is intended to provide candidate regions for review rather than pixel-accurate delineation. Deformable DETR is a suitable choice here because it preserves the end-to-end set-prediction formulation of DETR while improving convergence and small-object handling.

### A.5. Loss formulation

The model is trained with a joint objective:

$$\mathcal{L} = \mathcal{L}_{\text{cls}} + \lambda \mathcal{L}_{\text{det}},$$

where $\mathcal{L}_{\text{cls}}$ denotes the image-level classification loss and $\mathcal{L}_{\text{det}}$ denotes the detection loss. The classification loss is standard cross-entropy:

$$\mathcal{L}_{\text{cls}} = -\sum_{c=1}^{C} y_c \log \hat{p}_c.$$

The detection loss combines bipartite matching with box regression and objectness supervision:

$$\mathcal{L}_{\text{det}} = \mathcal{L}_{\text{box}} + \mathcal{L}_{\text{obj}}.$$

Here, $\mathcal{L}_{\text{box}}$ includes an $L_1$ term and a generalized IoU term, while $\mathcal{L}_{\text{obj}}$ is a binary loss on the objectness score.

## Appendix B. Datasets

We evaluated the model on two mammography datasets: OPTIMAM (OMI-DB) (Halling-Brown et al., 2020) and SGM1k, a biopsy-confirmed cohort from HCM Oncology Hospital. For both datasets, all images were preprocessed to retain only the breast region by cropping away the background and non-breast dark areas before model training and evaluation. Bounding-box annotations were then converted into the format required by the DETR-based framework, and all splits were performed at the patient level to avoid data leakage.

Table 2 summarizes the final dataset composition after preprocessing. OPTIMAM yielded 24,643 unique images from 7,851 patients, with 19,780 training images and 4,863 test images. SGM1k yielded 3,525 unique images from 1,002 patients, with 2,776 training images and 749 test images. The OPTIMAM cohort showed a lower proportion of images with multiple boxes but a higher maximum number of boxes per image, whereas SGM1k had fewer total images but a higher proportion of malignant cases.

## Appendix C. Experiments

All backbone variants were compared under a controlled experimental setting designed to isolate the effect of representation choice within the shared multi-task DETR architecture. Experiments were conducted on two mammography datasets, OPTIMAM (OMI-DB) and SGM1k; for both datasets, images were preprocessed by cropping to the breast region and removing background dark areas outside the breast, bounding-box annotations were converted to the DETR format, and all splits were performed at the patient level to prevent leakage. Input images were resized to $512 \times 512$, and each model was trained with a batch size of 32 for up to 400 epochs using a learning rate of $1 \times 10^{-4}$ and early stopping with a patience of 150 epochs. The decoder used 3 object queries and allowed at most 3 target objects per image during training. Image-level classification was optimized with focal loss, whereas the detection objective combined bounding-box regression, generalized IoU, and objectness

Table 2: Dataset statistics after preprocessing and patient-level splitting. All images were cropped to retain only the breast region.

| Statistic | OPTIMAM (OMI-DB) | | | Oncology Hospital (SGM1k) | | |
| --- | --- | --- | --- | --- | --- | --- |
| | Train | Test | Total | Train | Test | Total |
| Images (unique) | 19780 | 4863 | 24643 | 2776 | 749 | 3525 |
| Patients (unique) | 6332 | 1519 | 7851 | 802 | 200 | 1002 |
| Images with $> 1$ box | 1291 | 202 | 1493 | 424 | 88 | 512 |
| Maximum boxes/image | 16 | 9 | 16 | 4 | 4 | 4 |
| Benign | 12106 | 3056 | 15162 | 1135 | 319 | 1454 |
| Malignant | 7674 | 1807 | 9481 | 1641 | 430 | 2071 |

terms with weights $\lambda_{\text{bbox}}$=5.0, $\lambda_{\text{GIoU}}$=2.0, and $\lambda_{\text{obj}}$=1.0. Performance was evaluated using AUC, sensitivity, and specificity for classification, and mAP@.5 together with recall@.25 for lesion localization. Aside from backbone initialization, all training and evaluation settings were identical across experiments. Experiments were performed on a Linux server with 202.4 GB RAM and two NVIDIA L40 GPUs. Each L40 is an Ada Lovelace data-center GPU with 48 GB GDDR6 ECC memory and a 300 W maximum power rating.

## Appendix D. Grad-CAM Visualization

To provide qualitative insight into model behavior, we examined Grad-CAM visualizations for representative mammography cases across the evaluated backbones. Figure 2 shows that the stronger-performing backbones, particularly ConvNeXtV2 and DINOv3, more consistently concentrated attention on clinically relevant suspicious regions, whereas ResNet50 and MambaVision tended to produce less focused or less well-aligned responses in more difficult cases. These qualitative patterns are broadly consistent with the quantitative results in Table 1, where ConvNeXtV2 and DINOv3 achieved stronger overall classification and localization performance.

From an interpretability perspective, these maps should be viewed as supportive visual cues rather than definitive lesion localization. In mammography, exact lesion boundaries can be difficult to define because abnormalities are often small, subtle, and partially obscured by overlapping dense tissue. In this setting, approximate attention to suspicious regions may still be useful for highlighting candidate areas for review, even when the highlighted region does not precisely match the annotated box.

## Appendix E. Ethical Approval

Use of the OPTIMAM data in this study was conducted under the data access agreement dated 17/05/2023 between Cancer Research Horizons, Saigon Precision Medicine Research Center (SaigonMEC), and Royal Surrey NHS Foundation Trust. Use of the SGM1k cohort

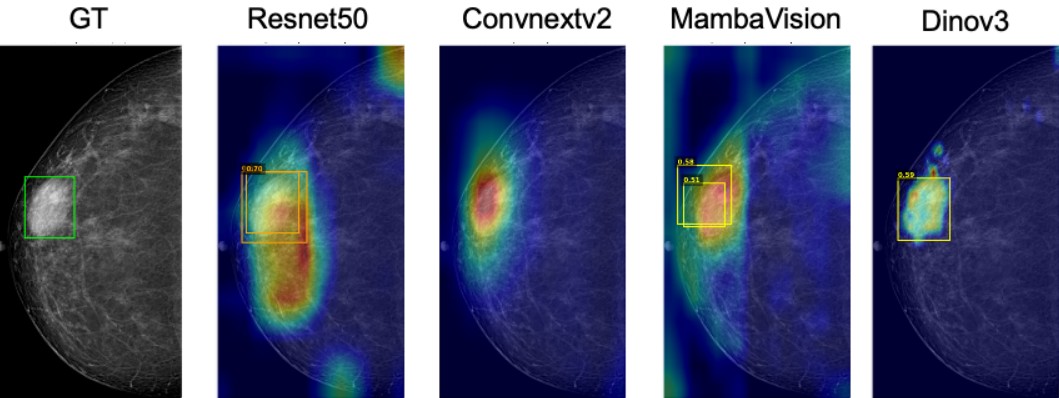

Figure 2: Qualitative comparison of Grad-CAM maps across backbones

was approved by Ho Chi Minh Oncology Hospital. All data use and analysis were performed in accordance with the relevant institutional approvals.

## Acknowledgments

The authors sincerely thank the doctors and staff at Pham Ngoc Thach University of Medicine for their valuable clinical and academic support. We also gratefully acknowledge the doctors and staff at Ho Chi Minh City Oncology Hospital for their assistance with clinical coordination and data-related activities. We further thank the student volunteers for their dedicated support in data collection and preparation, and the UTS eResearch team for their technical support and research infrastructure assistance.

