# OpenReview forum: "Modern Backbones Improve Multi-task DETR for Mammography Classification and Lesion Localization"
_MIDL.io/2026/Short_Papers — MIDL 2026 - Short Papers Poster_

### Official Review · Reviewer_cZEt · 2026-04-28
**Shared-encoder mammography classification and detection implemented with modern backbones**

**Rating:** 2
**Confidence:** 4

**Review:**

Please see "Strengths" and "Weaknesses" below.

**Summary:**

The authors evaluate malignancy classification and lesion localization with a shared encoder/backbone for mammography. They compare modern backbones (convnextv2, dinov3, mambavision) to an older backbone (resnet50) across two datasets, and find that performance is better when using the newer backbones.

**Strengths:**

- The paper is generally well-written, with a helpful "overview" figure to illustrate the framework.
- The task is implemented correctly, and architectures, loss, evaluation metrics, and other experimental design choices were chosen suitably.

**Weaknesses:**

- Overall, I think this is more of an "engineering" paper that just evaluated modern backbones for these specific mammography tasks. Essentially, all that the authors did was evaluate different backbones for the same two tasks (different prediction heads) on two datasets. As such, I don't see any clear unique contributions which the MIDL community would find useful; the main result is just that the newer backbones (convnext and dino) are more powerful than the older and smaller backbones (resnet), which is not surprising or unique. The tasks are not novel, and neither is using a shared encoder for both of them.
- One result is that DINOv3 > ConvNextv2 by performance on SGM1k, yet vice-versa on OPTIMAM. Yet, the authors do not discuss or attempt to explain this fairly interesting result.
- I'm also not sure of the purpose of the gradCAM example in the appendix; it's only shown for a single chosen image, so there are no clear findings to extrapolate from here. It seems kind of "thrown in" at the end.

**Justification Of Rating:**

MIDL short papers should briefly show interesting/novel preliminary results/findings that the medical image computing might find impactful, but this submission does not offer this. The authors essentially just implemented a few detection and classification (shared encoder) baselines using existing out-of-the-box architectures. I suggest that the authors determine if there are any unique, novel findings from this work beyond the unsurprising finding that modern, better backbones perform better than an older one.

---

### Decision · Program_Chairs · 2026-05-08

Accept (Poster)